# Multilingual Speech Recognition for Turkic Languages

Saida Mussakhojayeva, Kaisar Dauletbek, Rustem Yeshpanov and Huseyin Atakan Varol *

Institute of Smart Systems and Artificial Intelligence (ISSAI), Nazarbayev University, Astana 010000, Kazakhstan
* Correspondence: ahvarol@nu.edu.kz

**Abstract:** The primary aim of this study was to contribute to the development of multilingual automatic speech recognition for lower-resourced Turkic languages. Ten languages—Azerbaijani, Bashkir, Chuvash, Kazakh, Kyrgyz, Sakha, Tatar, Turkish, Uyghur, and Uzbek—were considered. A total of 22 models were developed (13 monolingual and 9 multilingual). The multilingual models that were trained using joint speech data performed more robustly than the baseline monolingual models, with the best model achieving an average character and word error rate reduction of 56.7%/54.3%, respectively. The results of the experiment showed that character and word error rate reduction was more likely when multilingual models were trained with data from related Turkic languages than when they were developed using data from unrelated, non-Turkic languages, such as English and Russian. The study also presented an open-source Turkish speech corpus. The corpus contains 218.2 h of transcribed speech with 186,171 utterances and is the largest publicly available Turkish dataset of its kind. The datasets and codes used to train the models are available for download from our GitHub page.

**Keywords:** automatic speech recognition; multilingual speech recognition; Turkic languages; transfer learning; Common Voice; big data; lower-resourced languages; Kazakh Speech Corpus; Uzbek Speech Corpus; Turkish Speech Corpus

## 1. Introduction

The task of automatic speech recognition (ASR) refers to converting any acoustic signal containing human speech into the corresponding word sequence [1]. The development of the graphics processing units (GPUs) and deep neural networks (DNNs) [2], the availability of transcribed speech corpora in the public domain [3–5], and the wide use of voice interaction services that support hundreds of languages (e.g., Alexa, Google Voice Assistant, and Siri) have led to ASR solutions achieving—and even exceeding—human performance [6]. That said, most ASR efforts have been directed towards developing models for languages for which large corpora exist (i.e., higher-resourced languages (The terms *lower-* and *higher-resourced languages* are used throughout the paper to emphasize the continuum existing across languages in terms of resources available for speech technology development.)), such as English, Mandarin, and Japanese (see, e.g., [4,7,8]). In turn, ASR models built for lower-resourced languages can rarely boast robustness and reliability due to the insufficient amount of training data.

To address the problem of the inadequacy of training data for lower-resourced languages, such techniques as transfer learning [9], data augmentation [10], and high resource transliteration [11], to name the most notable few, have been proposed. Special attention has also been devoted to the development of multilingual models, which enables the use of common linguistic features across languages, thus alleviating challenging data requirements [12]. Most work on multilingual ASR for lower-resourced languages focuses on combining the data of similar languages and performing cross-language optimization, by utilizing positive transfer from higher-resourced languages during training [13–15]. Research in transfer learning, too, has shown that linguistic similarity and relatedness generally lead to improved robustness of ASR models, particularly in resource-constrained

settings [16]. For example, linguistic relatedness and similarities have been made use of to build multilingual ASR models for lower-resourced Indian [17,18] and Ethiopian [19] languages and Arabic dialects [20]. The use of unrelated languages, however, generally results in a trade-off between quality and quantity, with models yielding performance comparable only with those of monolingual models [21] and with no significant improvement due to minimal linguistic overlap.

This work aims to make a contribution to the development of multilingual ASR for lower-resourced Turkic languages. To date, there have been studies conducted to develop multilingual models recognizing Turkic languages [21,22], but few Turkic languages were considered in the models or were recognized along with languages belonging to other language families (e.g., English, Persian, Russian, Swahili, etc.). In contrast, in this study, we exclusively focus on ten Turkic languages—namely, Azerbaijani, Bashkir, Chuvash, Kazakh, Kyrgyz, Sakha, Tatar, Turkish, Uyghur, and Uzbek.

According to various sources, the ten languages under consideration are at present spoken by 125–150 million speakers [23,24]. Spread over the vast area of Eurasia, the languages fall into several branches (see Table 1). With the exception of Chuvash and Sakha, which have peculiarities stemming from the early detachment from Common Turkic of the former and the influence of the Tungusic languages on the latter [23], the languages are, on the whole, remarkably similar in terms of lexis, phonology, and morphology. This is reflected in a certain degree of mutual intelligibility across the languages, with some of the most frequent words in the Turkic languages being exactly alike [25]. We therefore hypothesize that utilizing such features common for the ten languages is more likely to result in a robust multilingual ASR model than when unrelated languages are used, with some of the lower-resourced Turkic languages (e.g., Azerbaijani, Chuvash, and Sakha) benefiting from other Turkic languages for which more training resources are available (e.g., Bashkir, Kazakh, and Uzbek).

**Table 1.** The languages and datasets used in the study.

| Language | Code | Family | Branch | Script | Corpus | Validated Length (hr) | Utterances |
|---|---|---|---|---|---|---|---|
| Azerbaijani | az | Turkic | Oghuz | Latin | CVC | 0.13 | 81 |
| Bashkir | ba | Turkic | Kipchak | Cyrillic | CVC | 232.37 | 189,970 |
| Chuvash | ch | Turkic | Oghur | Cyrillic | CVC | 11.90 | 8651 |
| Kyrgyz | ky | Turkic | Kipchak | Cyrillic | CVC | 18.58 | 14,599 |
| Sakha | sa | Turkic | Siberian Turkic | Cyrillic | CVC | 6.61 | 3975 |
| Tatar | tt | Turkic | Kipchak | Cyrillic | CVC | 25.07 | 24,109 |
| Uyghur | ug | Turkic | Karluk | Arabic | CVC | 35.61 | 21,282 |
| Kazakh | kk | Turkic | Kipchak | Cyrillic | CVC | 1.60 | 1169 |
| | | | | | KSC | 332.60 | 153,853 |
| Turkish | tr | Turkic | Oghuz | Latin | CVC | 51.46 | 51,710 |
| | | | | | TSC | 218.24 | 186,171 |
| Uzbek | uz | Turkic | Karluk | Latin | CVC | 94.24 | 77,220 |
| | | | | | USC | 104.90 | 108,387 |
| English | en | Indo-European | West Germanic | Latin | CVC | 344.74 | 217,968 |
| Russian | ru | Indo-European | Slavic | Cyrillic | OpenSTT | 338.30 | 235,148 |

To contribute to the development of multilingual ASR for Turkic languages,

1. We compare the results of multilingual models trained on the data of the ten Turkic languages with the results of monolingual models trained for each of the languages;
2. We compare the results of the multilingual models with the results of models trained on the data of the ten Turkic languages and two non-Turkic languages (English and Russian);

3. We create the largest open-source speech corpus for the Turkish language that contains 218.2 h of transcribed speech.

The remainder of the paper is organized as follows: Section 2 provides an overview of existing work on multilingual ASR, focusing on both related and unrelated languages. In Section 3, we provide a description of the datasets used in the study and the procedures adopted to pre-process and split the data, as well as the details of the experimental setup. Section 4 describes the results obtained and a discussion of these results. Section 5 concludes the paper.

## 2. Related Work

The proliferation of studies in the field of ASR in recent years can be attributed to several factors, including a reduction in training time thanks to the use of the GPUs in deep learning [2], publicly available datasets (e.g., LibriSpeech [4] and DiDiSpeech [7]), and regular speech recognition competitions (e.g., CHiME-6 Challenge [26]). Demonstrating a significant performance boost [27–31], reporting a word error rate (WER) as low as 2–3% on popular datasets [32], and even achieving human parity [6], ASR research may create the false impression that the task is almost solved. However, the vast majority of the research focuses on mainstream languages for which extensive resources (e.g., recorded speech and human-labeled speech corpora) are available. For example, the whole Corpus of Spontaneous Japanese [8] contains a speech signal of about 661 h; the DiDiSpeech corpus of Mandarin [7] and the LibriSpeech corpus of read English speech [4] consist of about 800 and 1000 h of data, respectively. Consequently, languages that suffer from lower data availability can hardly afford the development of high-quality ASR systems.

One of the proposed ways to get around this problem is the application of transfer-learning techniques [33]. Even though the original idea explores reusing the weights of a previously trained DNN for a new task, it can be extrapolated to the problem of data insufficiency. In [9], the use of transfer learning in adapting a neural network originally trained for English ASR to German resulted in faster training, lower resource requirements, and reduced costs. Some studies propose similar methods, where the core idea is to train an ASR model jointly on multiple languages with the expectation that the system will perform better than systems trained on a single specific language. This approach is commonly referred to as multilingual ASR [34].

The earlier experiments with multilingual ASR [35,36] mostly explored the cases with only a few languages at a time and did not produce meaningful results except in language identification (LID) tasks. Language identifiers (IDs) are used as an additional input signal when multiple languages are involved, proving to be useful in both code-switching [37,38] and multilingual ASR [12,39]. There are two common ways to incorporate language IDs: (1) using special LID tokens at the beginning of output [37,38], thus using one-hot vector representation as an additional feature [12], or (2) using an auxiliary classifier in a multi-task setting [39].

Some recent advances in multilingual ASR assume that the presence of higher-resourced languages in the training set positively affects the performance of a model for lower-resourced languages [14,15,17–20,40]. In [14], the scholars showed that it is possible to train a massive single ASR architecture for 51 different languages and more than 16,000 h of speech across them, which, in practice, is significantly less time-consuming to tune than developing 51 individual monolingual baselines. It was also reported that training ASR multilingual models can improve recognition performance for all the languages involved, with the lower-resource languages observing a more significant reduction of WER and character error rate (CER) for East Asian languages. In another study [19] exploring ASR for lower-resourced languages, multilingual systems for four Ethiopian languages were developed. One of the models trained with speech data from 22 languages other than the target languages achieved a WER of 15.79%. Furthermore, the inclusion of the speech of a closely related language (in terms of phonetic overlap) in multilingual model training resulted in a relative WER reduction of 51.41%.

Most of the studies on multilingual ASR conclude that the average increase in performance produced by multilingual models, as opposed to monolingual ones, is higher for languages with greater linguistic overlap. Moreover, the development of a unified end-to-end (E2E) solution for a large number of languages that can potentially outperform monolingual models has become one of the focal points of multilingual ASR. However, research consistently shows that a model trained on a random set of languages does not consistently outperform monolingual models, even at a very large scale where more than 40 languages are used in the training set [12,14,15]. The authors of [14] have demonstrated that this is the case for higher-resourced languages, as the multilingual model failed to beat the baseline WER and CER scores for all higher-resourced settings.

This has led to the realization that using a dataset of languages with high linguistic overlap between them might yield better results. One of the ways to select these languages is to draw upon the language families to which they belong, as it is clear that the linguistic overlap between these languages is much greater than for languages with no inherent linguistic connections [19]. As a result, several recent studies into multilingual ASR have been carried out at the level of language families [17–20,40].

The authors of [18,20] developed E2E ASR systems for Indian and Arabic languages, respectively. Both papers report on average performance improvements over monolingual models, but were still unsuccessful in outperforming them in several languages. The findings were also consistent in the case of Ethiopian languages [19], where the scholars were able to obtain comparable results without having a target language in the training set. It is also important to note that the quality of training data may hinder the transfer learning capacity of the model, as was shown in [17]. The scholars were not able to achieve a significant improvement over monolingual experiments while using a dataset that contained systematic linguistic errors.

Most of the Turkic languages in our study are lower-resourced with few studies and datasets available. As can be seen from Table 1, these languages can be divided into five branches. Apart from Chuvash and Sakha, each belonging to a distinct subfamily, there are three major branches: Karluk, Kipchak, and Oghuz. To the best of our knowledge, while there are large open-source corpora for some of the languages belonging to the Karluk and Kipchak branches (e.g., the Bashkir set in Common Voice Corpus 10.0 (CVC) [3], Kazakh Speech Corpus (KSC) [41], and Uzbek Speech Corpus (USC) [42]), there are no similar or sufficiently large publicly available datasets for most of the languages under consideration. For example, in [43], a high-accuracy Tatar speech recognition system was trained on a proprietary dataset and the Tatar portion of CVC. Specifically, the model was trained on 328 h of unlabeled data and then finetuned on 129 h of annotated data, achieving a WER of 5.37% on the CVC test set. It should be noted that in this work, the ASR model was trained on a full Tatar CVC training set (28 h), which has 100% text overlap with the corresponding test set. Similarly, the authors of [44] developed an Uzbek ASR system trained on the Uzbek CVC (127 h) and Speechocean (https://en.speechocean.com/datacenter/details/1907.html (accessed on 22 January 2023)) (80 h) datasets and obtained a CER score of 5.41% on the Uzbek CVC test split. However, it is unclear whether the authors used part of the invalidated Uzbek CVC for training purposes, nor does the paper make mention of utterance overlap. In [45], different language models and acoustic training methodologies for the Azerbaijani language were investigated. Speech data of 80 h were collected from emergency calls. However, the data remain confidential, as they contain sensitive information about emergency cases.

As for the Turkish language, the corpus prepared by the Middle East Technical University (METU) [46,47] contains speech from 193 speakers (89 female and 104 male). Each speaker read 40 sentences that were selected randomly from a 2462-sentence set. Another Turkish speech corpus, containing broadcast news, was developed by Boğaziçi University [48] and has a total length of 194 h. The largest Turkish dataset [49] contains 350.27 h of validated speech. However, the data, which come from films and crowdsourcing, are

not publicly available. A detailed comparison between the existing Turkish ASR corpora and the Turkish Speech Corpus (TSC) can be found in Table 2.

**Table 2.** Turkish ASR datasets.

| Corpus | Length (hr) | Utterances | Open-Source |
| --- | --- | --- | --- |
| METU [46,47] | 5.6 | N/A | - |
| Boğaziçi [48] | 194 | N/A | - |
| HS [49] | 350.27 | 565,073 | - |
| CVC 10.0 [3] | 76 | 74,487 | + |
| TSC (ours) | 218.24 | 186,171 | + |

## 3. Materials and Methods

### 3.1. Data

#### 3.1.1. Datasets

To build a multilingual dataset, we considered a total of 12 languages. The ten Turkic languages were the target languages. English and Russian were the control languages. Detailed information on the languages utilized in this work is given in Table 1. As regards the datasets, we used multiple sources of transcribed speech, including the CVC [3], the Russian Open Speech To Text Dataset (OpenSTT) (https://github.com/snakers4/open_stt (accessed on 22 January 2023)), the KSC, the USC, and a new TSC.

The choice of the CVC as the main component of the multilingual dataset was due to the fact that it is one of the largest publicly available multilingual datasets designed for ASR purposes, comprising transcribed speech for 98 languages, including the target languages. The KSC is a large open-source corpus for Kazakh ASR, containing approximately 332 h of transcribed speech data comprising more than 153,000 utterances. The USC is the first open-source Uzbek speech dataset and comprises a total of 105 h of transcribed audio recordings by 958 different speakers.

With respect to the TSC, to the best of our knowledge, it is the largest Turkish speech corpus in the public domain. The data were collected from open sources and embraced various domains, such as news, interviews, talk shows, and documentaries. To acquire audio recordings, we used a command-line program to download videos from YouTube called *youtube-dl* (https://github.com/ytdl-org/youtube-dl (accessed on 22 January 2023)). . The resulting audio files were transcribed by a team of language specialists to ensure quality and accuracy. The TSC contains a total of 186,171 utterances, which adds up to 218.2 h of recorded speech. The total number of unique words is 122,319.

The data for the control languages, utilized to evaluate the relative performance of the multilingual models, came from the English and Russian subsets of the CVC and the OpenSTT, respectively, used in [21]. Specifically, the English data were 344 h in length and consisted of validated recordings that received the highest number of upvotes (i.e., instances of verification of correctness) from contributors. For evaluation purposes, we randomly extracted seven-hour subsets from the original CVC validation and test sets. For Russian, we used 338 h of speech, with associated transcripts previously corrected by native Russian speakers to ensure accuracy. The data embrace the domains of books and YouTube. A seven-hour subset of the data was selected for the development set. For the test set, we used the official validation sets of OpenSTT from both domains. Detailed statistics for each data source used in the study are given in Tables 1 and 3.

**Table 3.** Statistics for the training (Train), development (Dev), and test (Test) sets of CVC, OpenSTT , KSC, TSC, and USC.

| Language | Corpus | Length (hr) | | | Utterances | | |
|---|---|---|---|---|---|---|---|
| | | Train | Dev | Test | Train | Dev | Test |
| az | CVC | 0.05 | 0.04 | 0.04 | 39 | 20 | 22 |
| ba | CVC | 193.15 | 19.39 | 19.83 | 160,885 | 14,559 | 14,526 |
| ch | CVC | 8.47 | 1.54 | 1.89 | 6244 | 1140 | 1267 |
| ky | CVC | 14.25 | 2.14 | 2.19 | 11,373 | 1613 | 1613 |
| sa | CVC | 2.72 | 1.67 | 2.22 | 1643 | 1083 | 1249 |
| tt | CVC | 16.28 | 3.05 | 5.74 | 15,928 | 3062 | 5119 |
| ug | CVC | 26.27 | 4.54 | 4.80 | 15,787 | 2748 | 2747 |
| kk | CVC | 0.57 | 0.49 | 0.54 | 406 | 379 | 384 |
| | KSC | 318.40 | 7.13 | 7.07 | 147,326 | 3283 | 3334 |
| tr | CVC | 31.83 | 9.23 | 10.40 | 33,491 | 9095 | 9124 |
| | TSC | 209.6 | 4.26 | 4.38 | 179,259 | 3428 | 3484 |
| uz | CVC | 61.50 | 14.91 | 17.83 | 53,409 | 11,569 | 12,242 |
| | USC | 96.40 | 4.00 | 4.50 | 100,767 | 3783 | 3837 |
| en | CVC | 330 | 7.4 | 7.3 | 208,976 | 4346 | 4646 |
| ru | OpenSTT | 324.3 | 7.0 | 7.0 | 222,643 | 4776 | 7729 |

### 3.1.2. Data Pre-Processing

For each language, audio scripts were lowercased; character encodings were normalized, and punctuation marks were filtered out. For all the scripts (Arabic, Cyrillic, and Latin), we used the character-level encoding. We filtered the training data for each CVC dataset to keep the overlap between training and evaluation (development and test) sets below 40% to avoid memorization by the language models. The filtered utterances, as well as the script used to remove punctuation marks, can be found in our GitHub repository (https://github.com/IS2AI/TurkicASR (accessed on 22 January 2023)) for reproducibility purposes.

Four special tokens (i.e., $<blank>$, $<unk>$, $<space>$, and $<sos/eos>$) were used. One of the twelve language IDs ([az], [ba], [ch], [en], [kk], [ky], [ru], [sa], [tr], [tt], [ug], and [uz]) was prepended to each utterance. The character set size equaled 137 characters. Table 4 lists the 124 letters and symbols currently used in the alphabets of the languages under consideration. The remaining 13 characters (i.e., ئ, ا, ه و, قٖ, وٖ, ٷ, ٶ, ى, ي, ِ, â, î, ò, ŋ) can be encountered in words recorded using outdated or previous alphabets of the languages (e.g., Uyghur), as well as in loanwords or proper nouns containing symbols that do not originate in a particular language (e.g., diacritics). For reasons of efficiency, we removed utterances with more than 256 characters and audio recordings with a duration of more than 20 s.

**Table 4.** Characters of the considered languages.

| # | char | ug |
|---|------|----|
| 1 | ئا | + |
| 2 | ئە | + |
| 3 | ب | + |
| 4 | پ | + |
| 5 | ت | + |
| 6 | ج | + |
| 7 | چ | + |
| 8 | خ | + |
| 9 | د | + |
| 10 | ر | + |
| 11 | ز | + |
| 12 | ژ | + |
| 13 | س | + |
| 14 | ش | + |
| 15 | غ | + |
| 16 | ف | + |
| 17 | ق | + |
| 18 | ك | + |
| 19 | گ | + |
| 20 | ڭ | + |
| 21 | ل | + |
| 22 | م | + |
| 23 | ن | + |
| 24 | ھ | + |
| 25 | ئو | + |
| 26 | ئۇ | + |
| 27 | ئۆ | + |
| 28 | ئۈ | + |
| 29 | ۋ | + |
| 30 | ئې | + |
| 31 | ئى | + |
| 32 | ي | + |
| | | **32** |

| # | char | az | tr | uz | en |
|---|------|----|----|----|----|
| 1 | a | + | + | + | + |
| 2 | ə | + | - | - | - |
| 3 | b | + | + | + | + |
| 4 | c | + | + | - | + |
| 5 | ç | + | + | - | - |
| 6 | d | + | + | + | + |
| 7 | e | + | + | + | + |
| 8 | f | + | + | + | + |
| 9 | g | + | + | + | + |
| 10 | ğ | + | + | - | - |
| 11 | h | + | + | + | + |
| 12 | i | + | + | + | + |
| 13 | ı | + | + | - | - |
| 14 | j | + | + | + | + |
| 15 | k | + | + | + | + |
| 16 | l | + | + | + | + |
| 17 | m | + | + | + | + |
| 18 | n | + | + | + | + |
| 19 | o | + | + | + | + |
| 20 | ö | + | + | - | - |
| 21 | p | + | + | + | + |
| 22 | q | + | - | + | + |
| 23 | r | + | + | + | + |
| 24 | s | + | + | + | + |
| 25 | ş | + | + | - | - |
| 26 | t | + | + | + | + |
| 27 | u | + | + | + | + |
| 28 | ü | + | + | - | - |
| 29 | vs. | + | + | + | + |
| 30 | w | - | - | - | + |
| 31 | x | + | - | + | + |
| 32 | y | + | + | + | + |
| 33 | z | + | + | + | + |
| 34 | o' | - | - | + | - |
| 35 | g' | - | - | + | - |
| 36 | sh | - | - | + | - |
| 37 | ch | - | - | + | - |
| 38 | ng | - | - | + | - |
| 39 | ' | - | - | + | - |
| | | **32** | **29** | **30** | **26** |

| # | char | ba | ch | kk | ky | sa | tt | ru |
|---|------|----|----|----|----|----|----|----|
| 1 | а | + | + | + | + | + | + | + |
| 2 | ă | - | + | - | - | - | - | - |
| 3 | ә | + | - | + | - | - | + | - |
| 4 | б | + | + | + | + | + | + | + |
| 5 | в | + | + | + | + | + | + | + |
| 6 | г | + | + | + | + | + | + | + |
| 7 | ғ | + | - | + | - | - | - | - |
| 8 | ҕ | - | - | - | - | + | - | - |
| 9 | д | + | + | + | + | + | + | + |
| 10 | дь | - | - | - | - | + | - | - |
| 11 | е | + | + | + | + | + | + | + |
| 12 | ё | + | + | + | + | + | + | + |
| 13 | ĕ | - | + | - | - | - | - | - |
| 14 | ж | + | + | + | + | + | + | + |
| 15 | җ | - | - | - | - | - | + | - |
| 16 | з | + | + | + | + | + | + | + |
| 17 | ҙ | + | - | - | - | - | - | - |
| 18 | и | + | + | + | + | + | + | + |
| 19 | й | + | + | + | + | + | + | + |
| 20 | к | + | + | + | + | + | + | + |
| 21 | ҡ | + | - | - | - | - | - | - |
| 22 | қ | - | - | + | - | - | - | - |
| 23 | л | + | + | + | + | + | + | + |
| 24 | м | + | + | + | + | + | + | + |
| 25 | н | + | + | + | + | + | + | + |
| 26 | ң | + | - | + | + | - | + | - |
| 27 | ҥ | - | - | - | - | + | - | - |
| 28 | нь | - | - | - | - | + | - | - |
| 29 | о | + | + | + | + | + | + | + |
| 30 | ө | + | - | + | + | + | + | - |
| 31 | п | + | + | + | + | + | + | + |
| 32 | р | + | + | + | + | + | + | + |
| 33 | с | + | + | + | + | + | + | + |
| 34 | ç | + | + | - | - | - | - | - |
| 35 | т | + | + | + | + | + | + | + |
| 36 | у | + | + | + | + | + | + | + |
| 37 | ў | - | + | - | - | - | - | - |
| 38 | ұ | - | - | + | - | - | - | - |
| 39 | ү | + | - | + | + | + | + | - |
| 40 | ф | + | + | + | + | + | + | + |
| 41 | х | + | + | + | + | + | + | + |
| 42 | h | + | - | + | - | + | + | - |
| 43 | ц | + | + | + | + | + | + | + |
| 44 | ч | + | + | + | + | + | + | + |
| 45 | ш | + | + | + | + | + | + | + |
| 46 | щ | + | + | + | + | + | + | + |
| 47 | ъ | + | + | + | + | + | + | + |
| 48 | ы | + | + | + | + | + | + | + |
| 49 | і | - | - | + | - | - | - | - |
| 50 | ь | + | + | + | + | + | + | + |
| 51 | э | + | + | + | + | + | + | + |
| 52 | ю | + | + | + | + | + | + | + |
| 53 | я | + | + | + | + | + | + | + |
| | | **42** | **37** | **42** | **36** | **40** | **39** | **33** |

*Note.* The green and red shading indicates characters that are present in and do not belong to a specific language, respectively.

### 3.1.3. Data Augmentation

We applied speed perturbation [10] with factors of 0.9, 1.0, and 1.1 to the training sets. During training, we applied spectral augmentation [50] on-the-fly to the feature inputs of the encoder. Both data augmentation techniques are standard procedures regularly employed in ASR [17,20,21].

### 3.2. Experimental Setup

We first trained monolingual ASR models for each Turkic language on the CVC and then multilingual models. A total of 22 (13 monolingual and 9 multilingual) models were developed. A complete list of the models and the datasets on which they were trained can be found in Table 5. All of the models were trained on the training sets. Hyper-parameters were tuned using the development sets. The final models were evaluated on the test sets. Detailed information regarding the sets can be found in Table 3.

**Table 5.** A list of the models and the datasets used in training.

| | Model | | Corpus | | | | | | | | | | | | | | |
| | # | | CVC | | | | | | | | | | | Open STT | KSC | TSC | USC |
| Type | # | Name | az | ba | ch | kk | ky | sa | tt | tr | ug | uz | en | Open STT | KSC | TSC | USC |
|---|---|---|---|---|---|---|---|---|---|---|---|---|---|---|---|---|---|
| monolingual | 1 | az_cvc | + | - | - | - | - | - | - | - | - | - | - | - | - | - | - |
| | 2 | ba_cvc | - | + | - | - | - | - | - | - | - | - | - | - | - | - | - |
| | 3 | ch_cvc | - | - | + | - | - | - | - | - | - | - | - | - | - | - | - |
| | 4 | kk_cvc | - | - | - | + | - | - | - | - | - | - | - | - | - | - | - |
| | 5 | ky_cvc | - | - | - | - | + | - | - | - | - | - | - | - | - | - | - |
| | 6 | sa_cvc | - | - | - | - | - | + | - | - | - | - | - | - | - | - | - |
| | 7 | tt_cvc | - | - | - | - | - | - | + | - | - | - | - | - | - | - | - |
| | 8 | tr_cvc | - | - | - | - | - | - | - | + | - | - | - | - | - | - | - |
| | 9 | ug_cvc | - | - | - | - | - | - | - | - | + | - | - | - | - | - | - |
| | 10 | uz_cvc | - | - | - | - | - | - | - | - | - | + | - | - | - | - | - |
| | 11 | kk_ksc | - | - | - | - | - | - | - | - | - | - | - | - | + | - | - |
| | 12 | tr_tsc | - | - | - | - | - | - | - | - | - | - | - | - | - | + | - |
| | 13 | uz_usc | - | - | - | - | - | - | - | - | - | - | - | - | - | - | + |
| multilingual | 14 | turkic | + | + | + | + | + | + | + | + | + | + | - | - | - | - | - |
| | 15 | ksc_turkic | + | + | + | + | + | + | + | + | + | + | - | - | + | - | - |
| | 16 | tsc_turkic | + | + | + | + | + | + | + | + | + | + | - | - | - | + | - |
| | 17 | usc_turkic | + | + | + | + | + | + | + | + | + | + | - | - | - | - | + |
| | 18 | en_turkic | + | + | + | + | + | + | + | + | + | + | + | - | - | - | - |
| | 19 | ru_turkic | + | + | + | + | + | + | + | + | + | + | - | + | - | - | - |
| | 20 | en_ru_turkic | + | + | + | + | + | + | + | + | + | + | + | + | - | - | - |
| | 21 | all_turkic | + | + | + | + | + | + | + | + | + | + | - | - | + | + | + |
| | 22 | all_languages | + | + | + | + | + | + | + | + | + | + | + | + | + | + | + |

*Note.* The green shading indicates the datasets used in training a specific model.

### 3.2.1. Acoustic Models

We trained all models in Pytorch [51] using the ESPnet framework [52] and primarily followed the procedure described in the CVC recipe [52]. ESPnet is an end-to-end neural network toolkit that is a widely used open-source standard providing a complete setup for various speech processing tasks. We followed the latest conformer architecture of the CVC recipe when developing both monolingual and multilingual models. Specifically, the recipe for monolingual models is identical to the CVC recipe, while, for multilingual models, we increased the following hyperparameters:

- attention heads: $4 \rightarrow 8$
- encoder output dimension: $256 \rightarrow 512$
- convolutional kernel size: $15 \rightarrow 31$
- number of batch bins: $10^7 \rightarrow 4 \times 10^7$

These changes accommodated the increased amount of data used in training multilingual models and helped prevent overfitting.

Monolingual and multilingual models were trained on one and four NVIDIA DGX A100 (40 GB) GPUs, respectively. The combinations of datasets used for training the multilingual models are provided in Table 5. The monolingual model names are given

in the lowercase format `language code_dataset name` (e.g., `az_cvc` for the monolingual model trained on the Azerbaijani Common Voice Corpus). The multilingual model names are given in a lowercase format where

- `turkic` refers to models whose training data included the CVC datasets for the target Turkic languages,
- `ksc`, `tsc`, and `usc` refer to models whose training data included the Kazakh, Turkish, and Uzbek Speech Corpora, respectively,
- `en` and `ru` refer to models whose training data included the English and Russian datasets,
- `all_turkic` refers to the model whose training data included all datasets for the target Turkic languages,
- `all_languages` refers to the model whose training data included all datasets in the study.

As a baseline, the monolingual models were trained for each data source using the conformer architecture [27] with $42.98 \times 10^6$ trainable parameters. The decoder consisted of 6 transformer blocks, with a dropout rate set to 0.1. The same decoder configurations were also used for the multilingual architecture. For optimization, we used the Adam optimizer [53], with an initial learning rate of 4.0 and $2.5 \times 10^5$ warm-up steps.

In total, nine multilingual models were trained. All models had the same model configurations and were trained for 60 epochs based on the conformer architecture. We used 12 conformer encoder blocks with an output dimensionality of 512, 8 attention heads [30], a convolution kernel size of 31, and a dropout rate of 0.1. To optimize the training process, we used the Adam optimizer with the initial learning rate set at $25 \times 10^{-4}$ and $3 \times 10^5$ warm-up steps. The gradient clipping was set to 5 and the gradient accumulation to 4, while CTC [54] loss and label smoothing weights were set to 0.3 and 0.1, respectively. During inference, we used a beam size of 10 and set the CTC decoding weight to 0.6. The contribution weight of the language model was set to 0.3. All of the multilingual models had the same number of parameters, $108.68 \times 10^6$.

### 3.2.2. Language Models

For language models, we chose the transformer architecture [30]. We used sequential positional encoding, as the length of any utterance did not exceed 256 characters. Therefore, no sophisticated positional encoding methods were employed. Each of the language models had 16 transformer blocks, an embedding size of 128, with 8 attention heads, each with a dimensionality of 512. The dropout rate was set to 0.1 [55]. We trained the models for 30 epochs using the Adam optimizer. The initial learning rate was set to 0.001, with gradient clipping of 5.0 and gradient accumulation of 1. Similar to the acoustic models, we used $2.5 \times 10^4$ warm-up steps with batch bins set to $10^6$.

### 3.2.3. Performance Evaluation

The WER and CER metrics are the most common performance measures for ASR [56–58]. Even though the WER metric is usually preferred for most of the monolingual cases, calculating errors on the character level would convey the multilingual model performance in a more precise manner. The CER/WER of the predicted sequence is computed by dividing the sum of all substitutions, insertions, and deletions by the total number of characters/words in a reference transcription. The percentage of characters/words that have been inaccurately predicted is frequently related to CER/WER. However, CER/WER can exceed 100%, particularly when there are too many insertions. For example, the CER for a reference transcription 'fan' and a longer predicted sequence 'fantastic' is 200%, which is calculated by dividing the sum of substitutions (0), insertions (6 in 'tastic'), and deletions (0) in the predicted sequence ('fantastic') by the total number of characters in the reference transcription (3 in 'fan'). The performance of an ASR system improves as CER/WER decreases, with a value of 0% denoting the ideal result. In our study, CER/WER scores were transformed into percentages and displayed as such.

## 4. Results and Discussion

The performance of the models on the test sets is given in Tables 6 and 7. Considering the uncomparable distribution of data across the training, development, and test sets for some of the languages for which more than one dataset was available (i.e., Kazakh, Turkish, and Uzbek), we considered it fair and reasonable to evaluate the developed multilingual models separately on the CVC and the KSC, TSC, and USC test sets. While Table 6 provides the results obtained by the models on the CVC test sets exclusively, Table 7 contains the CER and WER scores for the models evaluated on the KSC, TSC, and USC test sets only. For readability, the dashed line separates the monolingual baselines from the multilingual models, and the green shading indicates the best results.

**Table 6.** The CER (%) | WER (%) results, average boost (AB, %) over the monolingual baseline, and training time (TT, day) of the models on the CVC test sets.

| Model | Language | | | | | | | | | | AB | TT |
|---|---|---|---|---|---|---|---|---|---|---|---|---|
| | az | ba | ch | kk | ky | sa | tt | tr | ug | uz | | |
| _cvc | 107.6│325.7 | 1.7│5.5 | 15.5│46.2 | 69.9│101.2 | 13.6│36.7 | 35.3│82.9 | 13.6│37.9 | 7.3│20.1 | 6.5│24.0 | 4.2│14.6 | - | - |
| turkic | 36.5│91.1 | 2.1│6.1 | 7.0│22.0 | 39.3│83.2 | 8.7│21.1 | 18.4│49.9 | 6.7│19.5 | 6.5│17.1 | 6.0│15.0 | 4.7│14.4 | 33.5│34.9 | 0.8 |
| ksc_turkic | 29.9│81.7 | 1.6│5.0 | 5.7│18.7 | 12.9│32.5 | 6.3│16.3 | 16.8│46.1 | 7.3│23.0 | 5.4│15.1 | 5.7│13.7 | 4.8│14.1 | 43.7│44.7 | 1.5 |
| tsc_turkic | 30.7│83.8 | 1.6│5.3 | 6.2│20.4 | 34.3│74.2 | 6.3│17.0 | 17.0│48.0 | 6.8│22.0 | 3.5│9.5 | 4.3│12.1 | 4.3│13.4 | 46.5│43.5 | 1.2 |
| usc_turkic | 34.0│86.4 | 1.8│5.8 | 6.8│21.8 | 37.7│84.0 | 7.7│18.9 | 18.6│50.2 | 5.8│17.6 | 5.6│15.2 | 4.8│12.6 | 3.9│12.3 | 41.6│39.7 | 1.1 |
| en_turkic | 34.9│85.3 | 1.7│5.3 | 6.0│18.9 | 36.8│81.6 | 7.8│18.7 | 19.6│50.9 | 6.2│18.6 | 5.9│14.7 | 6.2│15.8 | 5.5│14.7 | 35.8│38.4 | 1.7 |
| ru_turkic | 35.2│83.2 | 2.0│5.7 | 6.2│19.5 | 38.6│80.6 | 6.6│16.4 | 18.6│48.2 | 7.0│20.8 | 5.5│15.0 | 4.9│13.0 | 4.4│13.8 | 38.6│39.4 | 1.8 |
| en_ru_turkic | 31.5│85.3 | 1.6│5.1 | 5.7│18.0 | 32.0│75.0 | 6.2│16.6 | 17.9│49.4 | 6.0│18.6 | 5.7│16.1 | 5.2│13.7 | 4.6│13.8 | 42.3│40.8 | 3.2 |
| all_turkic | 26.7│75.9 | 1.5│4.9 | 4.9│17.2 | 11.7│29.0 | 4.9│13.1 | 15.7│45.0 | 5.6│18.1 | 3.3│9.0 | 4.1│11.0 | 3.0│10.3 | 56.7│54.3 | 2.4 |
| all_languages | 29.9│82.2 | 1.9│5.6 | 5.4│18.7 | 11.9│28.6 | 5.4│13.9 | 16.0│44.8 | 5.5│16.5 | 2.9│8.7 | 4.7│12.3 | 2.8│10.2 | 53.7│52.6 | 4.7 |

*Note.* The green shading indicates the best results.

**Table 7.** The CER (%) | WER (%) results of eight models on the KSC, TSC, and USC test sets.

| Model | Language | | |
|---|---|---|---|
| | kk | tr | uz |
| kk_ksc | 2.0│6.8 | - | - |
| tr_tsc | - | 3.8│12.6 | - |
| uz_usc | - | - | 5.0│16.8 |
| ksc_turkic | 1.5│5.7 | - | - |
| tsc_turkic | - | 2.9│9.6 | - |
| usc_turkic | - | - | 3.2│10.8 |
| all_turkic | 1.5│6.0 | 2.9│10.6 | 2.7│9.5 |
| all_languages | 1.5│5.9 | 3.0│10.8 | 2.9│10.2 |

*Note.* The green shading indicates the best results.

As can be seen from Table 6, for the CVC test sets, the `all_turkic` model, trained on the datasets of the Turkic languages, performed best, achieving the lowest CER and WER scores for six out of the ten target languages. The `all_languages` model, trained on all the 15 datasets in the study (with the addition of English and Russian), produced the lowest CER and WER scores for Tatar, Turkish, and Uzbek. Of note is Kazakh, for which the lowest CER score was achieved by `all_turkic`, while the lowest WER score was obtained by `all_languages`. However, the difference between the scores was negligibly small.

What stands out in Table 7 is that, when evaluated on the KSC, TSC, and USC test sets, the `all_turkic` and `all_languages` models mostly produced second best CER/WER scores, yielding to `ksc_turkic` and `tsc_turkic`, although not considerably. Nevertheless, `all_turkic` was able to achieve even lower CER/WER scores for Uzbek than `all_languages`, evaluated on the corresponding CVC test set.

### 4.1. Monolingual versus Multilingual Models

In Table 6, it is noticeable that all the monolingual models were outperformed by the multilingual models. To better illustrate how the ten Turkic languages were recognized by the monolingual models and the best performing `all_turkic` model, we present some of the decoded samples in Table 8.

From Table 6, we can see that improvement was at its peak for the lowest-resourced language in the study, Azerbaijani. With only a 0.13-hour-long dataset available, a significant CER/WER reduction from 107.6% to 26.7% and from 325.7% to 75.9%, respectively, was observed for this language. In Table 8, the monolingual az_cvc model appears to have output the same sequence *də* based on the likelihood model and thus did not produce correct results. In comparison, the all_turkic model generated both an intelligible and a comprehensible text with respect to the reference text, although it systematically failed to correctly predict words with the character ə, representing the /e/ sound, in the Azerbaijani utterance. Presumably, due to the lack of Azerbaijani training data, the model therefore proposed similar-sounding words that only slightly differed in spelling, originating from Turkish (*illerde*, *faaliyetine*) and Uzbek (*muxtalif*).

**Table 8.** Sample ASR results for monolingual models and the all_turkic model (R: reference, P_mono: prediction of a monolingual model, P_all_turkic: prediction of all_turkic).

| Lang | Type | Text | | | | | | | | CER | WER |
|------|------|------|------|------|------|------|------|------|------|-----|-----|
| **az** | R | *müxtəlif* | *illərdə* | *fərqli* | *sahələrdə* | *iş* | *fəaliyyətinə* | *başlayır* | | 0.0 | 0.0 |
| | P_mono | *\*\*\** | *də* | *də* | *də* | *də* | *də* | *ır* | | 171.7 | >100 |
| | P_all_turkic | *müxtalif* | *illerde* | *fergili* | *sohalarda* | *iş* | *faaliyetine* | *başlayır* | | 24.1 | 71.4 |
| **ba** | R | *лотерея* | *билеты* | *һыҙмаҡтыр* | *инде* | *ул* | | | | 0.0 | 0.0 |
| | P_mono | *нафария* | *пивиста* | *һыҙмаҡтыр* | *инде* | *ул* | | | | 30.3 | 33.3 |
| | P_all_turkic | *лотерея* | *билеты* | *һыҙмаҡтыр* | *инде* | *ул* | | | | 0.0 | 0.0 |
| **ch** | R | *заведующий* | *çemçe* | *диван* | *холодильник* | *микрохумлă* | *кăмака* | *ыйтнă* | | 0.0 | 0.0 |
| | P_mono | *хĕветувĕççи* | *çemçe* | *тиван* | *халакельн\*е* | *микра\*ăнлă* | *кăмата* | *ыйтнă* | | 38.3 | 100 |
| | P_all_turkic | *совету\*\*\** | *çemçe* | *тиван* | *холодильник* | *микрохумлă* | *камака* | *ыйтнă* | | 15.0 | 50.0 |
| **kk** | R | *өз* | *елімнің* | *басы* | *болмасам* | *да* | *сайыньıң* | *тасы* | *болайын* | 0.0 | 0.0 |
| | P_mono | *көзі жерген* | *жайдын* | *болан* | *боламан* | *жаланын* | *байдын* | *болан* | *байды* | 84.0 | >100 |
| | P_all_turkic | *үз* | *елемнің* | *басы* | *болмасам* | *да* | *сайыньıң* | *тасы* | *болайын* | 2.1 | 25.0 |
| **ky** | R | *исак* | *өзү* | *айткандай* | *аньıн* | *наньıн* | *бүт* | *көчөдөгүлөр* | *алчу* | 0.0 | 0.0 |
| | P_mono | *исак* | *өз* | *айткандай* | *аньıн* | *аньıн* | *бир* | *көчөдөгүлөр* | *болчу* | 73.1 | 50.0 |
| | P_all_turkic | *исак* | *өзү* | *айткандай* | *аньıн* | *нааньıн* | *бүт* | *көчөдөгүлөр* | *алчу* | 4.0 | 12.5 |
| **sa** | R | *оо* | *онтон* | *ону* | *баран* | *хахан* | *ылыахха* | *буоллаҕа* | *дии* | 0.0 | 0.0 |
| | P_mono | *оо* | *онтон* | *ому* | *байан* | *хахан* | *ынҕыах* | *тоҕуоллаҕа* | *диһи* | 19.1 | 75.0 |
| | P_all_turkic | *оо* | *онтон* | *уну* | *баран* | *хаан* | *ылыакка* | *булду* | *дии* | 17.8 | 47.6 |
| **tt** | R | *азанны* | *иң* | *оста* | *әйтүче* | *рөстәм* | *ибатуллин* | *булып* | *чыкты* | 0.0 | 0.0 |
| | P_mono | *узанны* | *иң* | *оста* | *итү* | *чәрстән* | *батыр* | *булып* | *чыкты* | 25.0 | 50.0 |
| | P_all_turkic | *азанны* | *иң* | *оста* | *итүче* | *рөстәм* | *ибатуллин* | *булып* | *чыкты* | 21.2 | 12.5 |
| **tr** | R | *ormanın* | *bütün* | *dalları* | *bütün* | *yaprakları* | *ötüyor* | *haykırıyordu* | | 0.0 | 0.0 |
| | P_mono | *ormanın* | *bütün* | *damları* | *bütün* | *yaprakları* | *atıyor* | *aykılıyordu* | | 10.0 | 42.9 |
| | P_all_turkic | *ormanın* | *bütün* | *dalları* | *bütün* | *yaprakları* | *ötüyor* | *haykırıyordu* | | 0.0 | 0.0 |
| **ug** | R | هببابلىناتتى | ماشىنا | ئبسل | شەھەرمەزد | چاغد | ئەينى | ماشىنسى | ئۇنىك | 0.0 | 0.0 |
| | P_mono | هببابلىناتتى | مۇشۇئى | شەئبسل | شەھەرمەزده | چاغدا | ئەينى | ماشىنسى | ئۇنىك | 14.1 | 62.5 |
| | P_all_turkic | هببابلىناتتى | ماشىنا | ئبسل | شەھەرمەزد | چاغد | ئەينى | ماشىنسى | ئۇنىك | 0.0 | 0.0 |
| **uz** | R | *biroq* | *o* | *sha* | *vaziyatda* | *bunga* | *jur* | *at* | *etolmadi* | 0.0 | 0.0 |
| | P_mono | *biroq* | *o* | *sha* | *vaziyatda* | *bundan* | *jur* | *at* | *etolmadim* | 6.7 | 25.0 |
| | P_all_turkic | *biroq* | *o* | *sha* | *vaziyatda* | *bundan* | *jur* | *at* | *etolmadi* | 4.7 | 12.5 |

*Note.* The green and red shading indicates correctly and incorrectly predicted words, respectively. A WER of >100 refers to the presence of insertion errors, which were not presented for visualization purposes. The asterisk signs (*) refer to deletion errors by a model.

The CER/WER reduction trend held for another two lower-resourced languages in the study. The multilingual models for Chuvash and Sakha—the only representatives of their branches—were able to notably decrease CER/WER for both languages, despite their considerable deviation from standard Turkic forms. The all_turkic model produced scores of 4.9%/17.2% and 15.7%/45.0% for Chuvash and Sakha, respectively, which is more than twice as low as the scores obtained by the corresponding monolingual models.

With respect to three Kipchak Turkic languages—namely, Bashkir, Kyrgyz, and Tatar—there was also a reduction in CER/WER observed, although to a different degree and thanks to different models. While the scores of 13.6%/37.9% by the Tatar monolingual model were reduced to 5.5%/16.5% by all_languages, it was all_turkic again that took the Kyrgyz baseline scores down to 4.9%/13.1%. That said, the monolingual model for Bashkir—the Turkic language whose CVC data were over 230 h in length—yielded CER/WER scores that were not considerably higher than the lowest scores by all_turkic, 1.7%/5.5% and 1.5%/4.9%, respectively. These observations seem to suggest that CER/WER reduction is

more notable for languages with lower amounts of (CVC) data (e.g., Azerbaijani, Chuvash, Kazakh, and Sakha) and less evident for languages with a higher number of resources (e.g., Bashkir and Uzbek). Despite the less remarkable CER/WER improvement for Bashkir than for the lower-resourced languages, it can be clearly seen in Table 8 that, in contrast to the monolingual `ba_cvc` model, the `all_turkic` model was successful in recognizing loanwords, especially those taken from Russian and instantly familiar to most people in the former Soviet countries (*лотерея*, *билеты*). Similarly, the `all_turkic` model outperformed the monolingual Chuvash model in predicting loanwords, recognizing some completely correctly (*холодильник*) and others to varying degrees (*заведующий* → *совету*\*\*\*, *диван* → *тиван*).

ASR for Uyghur, a language of the Karluk branch, also seems to have notably benefited from the development of multilingual models. One can see a steady decrement in CER/WER as the data of other Turkic languages were added to the training set. The joint use of data of all the Turkic languages in the `all_turkic` model resulted in scores of 4.1%/11.0%.

In the case of Kazakh, Turkish, and Uzbek—the three languages in the study for which in addition to the CVC there was another speech corpus used for model development—the data in Tables 6 and 7 appear to suggest that the results may vary depending on the training and test sets used. To begin with, the Kazakh and Turkish monolingual models trained on the CVC data produced notably higher CER/WER results than the monolingual models trained on the KSC and the TSC. This can probably be attributed to the marked difference in the size of the training data. It is especially the case for Kazakh, for which the total amount of the CVC data was as little as 1.60 h as opposed to the hefty 332.60 h in the KSC. Thus, it seems nothing but expected that `kk_ksc` and `tr_tsc` achieved the remarkable 2.0%/6.8% and 3.8%/12.6%, respectively, as compared to the 69.9%/101.2% of `kk_cvc` and the 7.3%/20.1% of `tr_cvc`. For example, the scores of `uz_cvc` and `uz_usc` were quite similar—although slightly lower for the former (4.2%/14.6% and 5.0%/16.8%, respectively), for the two Uzbek datasets were comparable in size.

As regards the multilingual models for Kazakh, Turkish, and Uzbek, when evaluated on the CVC test sets, the best performance was achieved by the `all_languages` model. While the CER score for Turkish and Uzbek was approximately 2.9%, the WER score held in the range of 8.7% to 10.2%. For Kazakh, the model produced the lowest WER score (28.6%), but achieved the second best CER score of 11.9%, yielding to `all_turkic` with 11.7%.

On the other hand, in the evaluation of the multilingual models on the KSC, TSC, and USC test sets, the best CER and WER results of 1.5% and 5.7%, respectively, in Kazakh ASR were produced by the `ksc_turkic` model. Such low scores are likely to have been achieved owing to the sufficient amount of data in the training and test sets for the model to learn from and test its hypotheses on. The CER scores produced by `all_turkic` and `all_languages` were identical to that of `ksc_turkic`, with the WER scores being only negligibly higher. For Turkish, the lowest scores were achieved by `tsc_turkic`, 2.9%/9.6%. Although the model exhibited a CER result lower than that obtained on the CVC test set, the WER score was still slightly higher. Looking at the scores for Turkish ASR in Tables 6 and 7, it is apparent that the multilingual models evaluated both on the CVC and TSC test sets produced somewhat similar results. In the case of the Uzbek language, the result of 2.7%/9.5% achieved by `all_turkic` was the lowest in the evaluation of the multilingual models on both test sets.

### 4.2. Multilingual ASR versus Transfer Learning

For comparison purposes, we conducted additional experiments using transfer learning. We pre-trained monolingual models for the two highest-resourced Turkic languages in the study (i.e., Kazakh and Turkish). Then, we finetuned the models on the three lowest-resourced Turkic languages (i.e., Azerbaijani, Chuvash, and Sakha).

As can be seen from Table 9, the two models built using transfer learning (`tl_ksc` and `tl_tsc`) produced considerably lower CER/WER scores than those of the monolingual baselines. That said, they were still higher than the scores of the multilingual `all_turkic`

model. Of note was the CER score for Sakha produced by the model that was pre-trained on Kazakh data, proving the best CER score in the study for this language.

**Table 9.** The CER (%) | WER (%) results of monolingual models, models built using transfer learning, and `all_turkic` for Azerbaijani, Chuvash, and Sakha.

| Model | Model Type | Language | | |
|---|---|---|---|---|
| | | **az** | **ch** | **sa** |
| `_cvc` | monolingual baseline | 107.6 | 325.7 | 15.5 | 46.2 | 35.3 | 82.9 |
| `tl_ksc` | transfer learning | 87.1 | 329.8 | 7.6 | 30.2 | 15.3 | 54.6 |
| `tl_tsc` | transfer learning | 86.7 | 290.1 | 8.4 | 31.7 | 17.4 | 59.1 |
| `all_turkic` | multilingual | 26.7 | 75.9 | 4.9 | 17.2 | 15.7 | 45.0 |

*Note.* The green shading indicates the best results.

### 4.3. Turkic versus Non-Turkic

The experiments clearly show that ASR for the Turkic languages appears to have benefited more from multilingual models trained jointly on the data of other related (Turkic) languages (e.g., `all_turkic`) than from models developed using data from non-Turkic (control) languages (i.e., English and Russian). We attribute this to essential linguistic features shared by Turkic languages [23,24].

Nevertheless, looking at Table 6, one cannot but admit that the CER/WER scores for six out of the ten Turkic languages produced by the `en_ru_turkic` model are appealing. The joint use of English and Russian training data led to a remarkable CER/WER reduction from 107.6%/325.7% to 31.5%/85.3.6% for Azerbaijani and an approximately twofold CER/WER decrease for Chuvash, Kazakh, Kyrgyz, Sakha, and Tatar. Of note are also the results of `ru_turkic` for Turkish and Uyghur. Although the scores were higher than those of `all_turkic` and `all_languages`, they were still lower than those of the corresponding monolingual baselines.

While the results of `en_ru_turkic` can be attributed to the likely presence of international words found (phonetically almost unchanged) in many languages (e.g., alcohol, Internet, computer, etc.) and Russian loanwords widely used in the six languages, the reduction in CER/WER for Turkish and Uyghur could be explained by the findings of [59]. The researchers found that the amount of source language data was more important than the relatedness of the source language to the target language, yielding greater performance. In other words, training a model on more than 300 h of transcribed speech in an unrelated language is more likely to result in CER/WER reduction than developing a model trained on data of a related and similar language, but of a smaller size.

### 4.4. Language Identification

Since we prepended language IDs to utterances, we were also able to evaluate the best-performing model, `all_turkic`, in terms of LID. The confusion matrix in Table 10 provides a clearer insight into the model performance and the errors made in predicting the language of an utterance. These results were obtained on all the target language test sets, which included the CVC, the KSC, the TSC, and the USC.

As can be seen from Table 10, the accuracy of the `all_turkic` model in LID was above 97% for seven out the ten Turkic languages. While the LID accuracy scores of 36.36% for Azerbaijani and 77.50% for Sakha may be explained by the insufficient amount of training data available for the languages (only 6.74 h in aggregate), the score of 80.86% for Tatar is the result of the frequent failure of the model to discriminate Tatar from Bashkir. Almost 16% of the Tatar utterances were identified as Bashkir, which should come as no surprise, given the close phonetic affinity of the languages, which differ mainly in their consonant systems [24]. When failing to unambiguously identify the language, this reliance of the model on phonetic similarities between languages—being particularly strong when they belong to the same branch—can be especially observed for Azerbaijani, Kyrgyz, and

Uyghur. Of 22 Azerbaijani utterances, ten were predicted to be Turkish, both languages being from the Oghuz branch. Most of the erroneously predicted Kyrgyz utterances were identified as either Bashkir or Kazakh (Kipchak languages). The second most likely language in the recognition of Uyghur utterances was Uzbek, the other language from the Karluk branch.

However, this should be taken with a grain of salt, for this observation also holds when the amount of training data of the actual language is lower than that of the closely related but falsely predicted language. That is, we can assume that Azerbaijani utterances were often misidentified as Turkish, Tatar utterances as Bashkir, and Uyghur speech as Uzbek, mainly because there were fewer training data for Azerbaijani, Tatar, and Uyghur than for their close relatives. A closer look at Table 10 reveals that Turkish, Bashkir, and Uzbek, when misidentified, were not necessarily recognized as languages that come from the same branch, but rather as languages with data of considerable size in the training set. The case of Sakha—the second lowest-resourced and one of the two languages with the greatest linguistic distance from the other languages in the study, the other being Chuvash—can serve as an example. It is apparent from the confusion matrix that the `all_turkic` model only minimally confused the other languages with Sakha. Overall, we can conclude that the amount of data of a language in the mixed dataset and language relatedness were probably the two most important factors influencing the ability of the `all_turkic` model to successfully identify languages.

**Table 10.** Language identification confusion matrix and accuracy (Acc, %) of the `all_turkic` model for different Turkic languages on the combined test sets (CVC, KSC, TSC, and USC).

| | | Predicted Language | | | | | | | | | | |
|---|---|---|---|---|---|---|---|---|---|---|---|---|
| | **Lang** | **az** | **ba** | **ch** | **kk** | **ky** | **sa** | **tr** | **tt** | **ug** | **uz** | **Acc** |
| **Actual Language** | **az** | 8 | 2 | 0 | 0 | 0 | 0 | 10 | 0 | 0 | 2 | 36.36 |
| | **ba** | 0 | 14,400 | 12 | 2 | 15 | 0 | 43 | 8 | 18 | 28 | 99.13 |
| | **ch** | 0 | 7 | 1,241 | 0 | 4 | 0 | 8 | 7 | 0 | 0 | 97.95 |
| | **kk** | 0 | 34 | 6 | 3634 | 23 | 0 | 2 | 0 | 8 | 11 | 97.74 |
| | **ky** | 2 | 17 | 11 | 22 | 1534 | 2 | 6 | 12 | 2 | 5 | 95.10 |
| | **sa** | 4 | 108 | 10 | 8 | 105 | 968 | 2 | 25 | 11 | 8 | 77.50 |
| | **tr** | 34 | 181 | 26 | 4 | 25 | 1 | 12,242 | 12 | 39 | 46 | 97.08 |
| | **tt** | 2 | 817 | 41 | 30 | 25 | 2 | 22 | 4139 | 12 | 29 | 80.86 |
| | **ug** | 1 | 9 | 2 | 1 | 17 | 0 | 2 | 2 | 2681 | 32 | 97.60 |
| | **uz** | 13 | 133 | 10 | 3 | 17 | 0 | 18 | 15 | 26 | 15,833 | 98.54 |

*Note.* The green shading indicates the instances where an actual class and a predicted class match.

## 5. Conclusions

This study set out to develop a multilingual ASR model for lower-resourced Turkic languages. Ten languages—namely, Azerbaijani, Bashkir, Chuvash, Kazakh, Kyrgyz, Sakha, Tatar, Turkish, Uyghur, and Uzbek—were considered. A total of 22 models were developed, of which 13 were monolingual and 9 multilingual. The multilingual models outperformed the monolingual baselines, with the best performing model (i.e., `all_turkic`) achieving an average boost of 56.7%/54.3% in CER and WER reduction, respectively, for six out of the ten languages. The experiment results showed that CER and WER reduction was more likely to be observed when multilingual models were trained on the data of Turkic languages than when developed using data from such non-Turkic languages as English and Russian. The study also presented the TSC—an open-source speech corpus for the Turkish language. The corpus contains 218.2 h of transcribed speech comprising over 186,171 utterances and is the largest publicly available Turkish dataset of its kind. The datasets and codes used to train the models are available for download from https://github.com/IS2AI/TurkicASR (accessed on 22 January 2023). It is hoped that our work will stimulate further efforts in training ASR systems for Turkic languages. Presumably, the `all_turkic` model can serve as a springboard for transfer learning for monolingual ASR models for other Turkic

languages whose inclusion in our study proved challenging due to the lack of open-source corpora.

**Author Contributions:** Conceptualization, S.M. and H.A.V.; methodology, K.D., R.Y. and H.A.V.; software, K.D. and S.M.; validation, K.D. and S.M.; formal analysis, R.Y. and K.D.; investigation, S.M., R.Y. and K.D.; resources, H.A.V.; data curation, S.M. and H.A.V.; writing—original draft preparation, K.D., S.M. and R.Y.; writing—review and editing, R.Y. and H.A.V.; visualization, K.D.; supervision, H.A.V.; project administration, S.M.; funding acquisition, H.A.V. All authors have read and agreed to the published version of the manuscript.

**Funding:** This research received no external funding.

**Informed Consent Statement:** Not applicable.

**Data Availability Statement:** Publicly available datasets were analyzed in this study. The datasets and models can be found here: https://github.com/IS2AI/TurkicASR (accessed on 22 January 2023).

**Conflicts of Interest:** The authors declare no conflict of interest.

## Abbreviations

The following abbreviations are used in this manuscript:

| | |
|---|---|
| CER | character error rate |
| CTC | connectionist temporal classification |
| CVC | Common Voice Corpus 10.0 |
| DNN | deep neural network |
| E2E | end-to-end |
| GPU | graphics processing unit |
| ID | identifier |
| KSC | Kazakh Speech Corpus |
| LID | language identification |
| METU | the Middle East Technical University |
| OpenSTT | Russian Open Speech To Text Dataset |
| TSC | Turkish Speech Corpus |
| USC | Uzbek Speech Corpus |
| WER | word error rate |

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
