# Peer review of "Multilingual Speech Recognition for Turkic Languages"

_information, doi:10.3390/info14020074_

Round 1

Reviewer 1 Report

[Comment 1] Novelty

[Subcomment 1a] (lines 188-190) The explanations here seems to show that the authors generated the database by themselves, instead of only using any existing one. If it is true, the authors should state it clearly in line 69.

[Subcomment 1b] The authors must compare the information of the generated data with the existing datasets, e.g., in a table.

[Comment 2] Data generation

(lines 218-220) How did the authors select the characters to remove? Such an explanation is important to allow next researchers to reproduce the data generation scheme.

[Comment 3] References

(line 250) Please provide a reason or reference that shows why the authors used such kernel size, dropout rate, and batch bins. The authors need to do the same for any other defined parameters that have no reference yet.

[Comment 4] Writing quality and clarity

[Subcomment 4a] (lines 47-48) It is difficult to understand the meaning of this part of sentence: "...they were recognized among other - non-Turkic- languages." Please revise the sentence or provide more explanations.

[Subcomment 4b] When presenting the multilingual model names, the authors must provide the long form for each of them, for clarity, e.g., what is "trkc"?

[Subcomment 4c] (lines 273-274) The sentence could not be understood easily. Please provide at least one example (with CER/WER below 0% or more than 100%).

[Subcomment 4d] Because the authors use the abbreviations a lot, please provide the list of abbreviations at the earlier part of the manuscript to allow ease search for the abbreviations.

Author Response

Please see the attached pdf file.

Reviewer 2 Report

The article repeats a lot of descriptive information in the opening chapters and does not bring many new findings. It just confirms the conclusions of other studies that have focused on other groups of similar or related languages.

I therefore recommend to perform additional experiments. For example, focused on the use of transfer learning - it would be convenient to train a model for a smaller language using the Turkish model for initialization. This approach could lead to better results than multilingual training.

One of the main contributions of the article is the creation of an open-source corpus for Turkish. However, the link from github leads to a google form, which cannot be displayed due to limited permissions:

https://docs.google.com/forms/d/e/1FAIpQLSeqOficzzzIEEnJU4Am-JBdty3V6rYERtE2mv5mVD1WpiOZkw/formrestricted

This fact contradicts the authors' claim that the corpus is open-source.

Author Response

Please see attached pdf file.

Round 2

Reviewer 1 Report

Thank you for your revisions.

Reviewer 2 Report

The authors has improved the paper according to my suggestions.